

# A retrospective study comparing super-mini percutaneous nephrolithotomy and flexible ureteroscopy for the treatment of 20–30 mm renal stones in obese patients

Chen Xu[1,*], Rijin Song[2,*], Pei Lu[2], Minjun Jiang[1], Guohua Zeng[3] and Wei Zhang[2]

[1] Urology, The Ninth People's Hospital of Suzhou City, Suzhou, China
[2] Urology, The First Affiliated Hospital with Nanjing Medical University, Nanjing, China
[3] Guangdong Key Laboratory of Urology, The First Affiliated Hospital of Guangzhou Medical University, Guangzhou, China
[*] These authors contributed equally to this work.

Corresponding author
Wei Zhang, weizhangsir@126.com

## ABSTRACT

**Objective**. This study aimed to compare the efficacy and safety of Super-mini percutaneous nephrolithotomy (SMP) and flexible ureteroscopy (F-URS) in the treatment of 20–30 mm renal stones in obese patients.

**Methods**. We conducted a retrospective analysis of outcomes of patients who underwent SMP and F-URS to treat 20–30 mm renal stones from August 2017 to September 2018. Patients with BMI $>30$ kg/m$^2$ were enrolled into this study. Forty-eight patients underwent SMP, while 104 patients underwent F-URS by the same surgeon. The patients' demographic data, stone characteristics, perioperative parameters and outcomes, complications, stone-free rate (SFR) and overall costs were retrospectively assessed.

**Results**. No significant differences were found between the two groups in terms of age, gender, BMI, operation side, stone size, number, locations, stone compositions and CT value. The mean operation time was significantly shorter in the SMP group ($p < 0.001$), while the F-URS group had significantly shorter postoperative stays ($p < 0.001$) and lower complication rates ($p < 0.001$). Both groups had similar SFR at a 3-month follow-up ($p = 0.190$), while the SMP group achieved significant higher SFR 3 days after the operation ($p < 0.001$). The SMP group had a significantly lower overall cost and fewer stage-2 procedures than the F-URS group.

**Conclusion**. SMP and F-URS are equally effective in obese patients with 20–30 mm renal stones. However, F-URS offers the advantage of a lower complication rate, while SMP performed better in terms of operation time, tubeless rate, stage-2 procedures and overall costs.

## INTRODUCTION

Obesity is the world's most prevalent health problem and a leading cause of preventable death. There has been a significant increase in the rate of obesity during the last two decades, a result of our increasingly sedentary lifestyles and high-fat diets (*Krzysztoszek, Wierzejska & Zielinska, 2015*). It is well known that obese patients suffer a higher risk of nephrolithiasis (*Wong, Cook & Somani, 2015*; *Traxer et al., 2008*). Since discovering this, tremendous progress has been made in understanding the etiology, the diagnosis and particularly the treatment of nephrolithiasis in obese patients.

With the development of endoscopic technology and the growing use of endoscopic surgery, minimally invasive treatment modalities such as flexible ureteroscopy (F-URS) and percutaneous nephrolithotomy (PCNL) are being widely used for large renal stones and has resulted in acceptable stone free and complication rates. Several studies recommend F-URS for its high success and low complication rates when dealing with renal stones ≥20 mm (*Cohen, Cohen & Grasso, 2013*; *Doizi et al., 2015*; *Xu et al., 2015*). Nevertheless, the use of F-URS in obese patients presents challenges including long-term stone discharge, increased overall cost and requirement of multiple interventions (*Sari et al., 2013*). However, traditional PCNL has been proven to be a safe and effective alternative in obese patients (*Trudeau et al., 2016*; *Zhou et al., 2017*). When compared with conventional PCNL, super-mini percutaneous nephrolithotomy (SMP) causes less blood loss and fewer perioperative complications (*Zeng et al., 2016*; *Liu et al., 2018*). However, it still remains unknown whether F-URS or SMP would be a better choice for patients with body mass indexes (BMIs) >30 kg/m$^2$. In this study, we aimed to evaluate and compare the efficacy and safety of F-URS and SMP for the treatment of 20–30 mm renal stones in obese patients.

## MATERIALS AND METHODS

A total of 879 patients that underwent F-URS or SMP for renal stones at our center from August 2017 to September 2018 were analyzed. 152 obese patients from this period had 20–30 mm renal stones and BMIs >30 kg/m$^2$. We selected these 152 patients for our study. A total of 104 of these patients underwent F-URS, while 48 received SMP. All procedures were performed by the same senior surgeon (Pro W Zhang). The surgical procedure was chosen by each patient, and they all provided written informed consent before their operation. Necessary preoperative diagnostic procedures (detailed medical history, physical examination, blood tests, serum creatinine, urine tests, sterile urine culture, 24-hour urine electrolytes and parathyroid hormone) were performed in all patients. Renal stones and kidney characteristics were evaluated through plain X-rays of kidney-ureter-bladder (KUB) and low-dose abdominal computed tomography (CT). Patients younger than 18 years old, patients diagnosed with hyperparathyroidism and patients with renal abnormalities were excluded from our study. Data analysis included patients' demographics, stone characteristics, surgical details, perioperative outcomes and SFR. The study protocol was approved by the ethics committee of the Ninth Peoples' Hospital of Suzhou city (approval number (19-01)).

## Surgical technique

### F-URS technique

All F-URS patients underwent a standardized F-URS by the same surgeon using a 7.95- Fr flexible ureteroscope (Olympus, URFP6, Japan). These patients were placed in the dorsal lithotomy position under general anesthesia, and intravenous antibiotics (according to their preoperative urine culture) were administered 30 min preceding each operation. Ureteroscopy was routinely performed using a semi-rigid ureteroscope (8F-9.8F, Richard Wolf GmbH, Knittlingen, Germany) in order to place a guide wire (Sensor) into the target renal pelvis and for visual assessment of the ureter, after which a hydrophilic-coated ureteral access sheath (UAS;12-14F;COOK Medical; Ireland) was inserted alongside the guide wire. If the UAS could not be placed due to ureteral stenosis or kinking, a 7- Fr double-J stent was inserted, and the F-URS was performed 2 weeks later. Renal stones were treated using a 365 um holmium laser with energy of 0.8–1.0 J and a rate of 10–30 Hz, based on the stone volume. Stone fragments ≤2 mm were left for spontaneous passage, while basket retrieval was used for stones larger than 2 mm. A 6-Fr double-J stent was left in place in all cases. Stones were collected postoperatively for further analysis through infrared spectroscopy. Operation time was defined as the time passed from the insertion of the semi-rigid ureteroscope to the completion of the double-J stent placement. All patients underwent a kidney-ureter-bladder (KUB) radiography 3 days after the surgery to evaluate the primary SFR. For patients with significant residual stones, a subsequent F-URS procedure was performed 2 weeks later. The double-J stent was usually removed within 4 weeks following the operation.

### SMP technique

All SMP patients were placed in the lithotomy position under general anesthesia, with a 6-Fr open end ureteral catheter inserted up to the operative collecting system. Then, the patient was turned into the prone position. B-ultrasound was used to assess the calyceal system configuration, the stone positions and the route of puncture. The target calyx was precisely punctured by a needle under ultrasound guidance, then a 0.032-in J guide wire was placed into the selected pelvicaliceal system. Dilation was performed using serial fascial dilators until a 12/14 evacuation sheath was inserted in the tract. A suction-evacuation sheath of the corresponding size was then placed and connected to a specimen collection bottle with an attached negative pressure aspirator. With the help of a miniature endoscope, the renal stones were fragmented using a 365-um holmium laser. Finally, a double-J stent was only placed in the following cases: an obstructing inflammatory ureteral edema, evidence of pelvic-ureteric junction obstruction, significant pyelocaliceal blood clots after the lithotripsy, concurrent treatment of ipsilateral ureteral stones with semi-rigid ureteroscopy patients with residual stones. The sheath was removed, and the wound was sutured at the end of all procedures. For patients with significant residual stone fragments who required a second-stage procedure or if there was significant bleeding or extravasation, a nephrostomy tube was necessary. Operation time was calculated from the time the semi-rigid ureteroscope was inserted to the completion of the wound suture or the nephrostomy tube placement. A KUB radiogram was performed 3 days postoperatively to

assess the primary SFR. When faced with significant residual stone fragments, an additional procedure was performed 4 weeks later. Otherwise, the nephrostomy tube (if inserted) was clamped and removed immediately. The double-J stent (if inserted) was always removed 4 weeks postoperatively through cystoscopy under local anesthesia.

All patients received follow-up KUB between 1 and 3 months after their operations during outpatient care. This procedure re-assessed the SFR. We defined stone-free status as the absence of any stone or stone fragment ≤2 mm in the kidney. Postoperative complications were graded according to the Clavien-Dindo classification system (*Dindo, Demartines & Clavien, 2004*).

### Statistical analysis

Statistical analysis was performed with a SPSS statistics package, version 20.0. The data were expressed as mean ± standard deviation (SD). Statistical analysis was carried out using the Chi-square test for categorical variables and the Student's $t$-test for continuous variables. $P < 0.05$ was considered statistically significant.

## RESULTS

A total of 152 obese patients with 20–30 mm renal stones were enrolled in this study. The patients' demographic and stone characteristics are outlined in Table 1. No significant differences were found between the two groups for age, gender, BMI and operation side ($p = 0.610$, $p = 0.751$, $p = 0.458$, $p = 0.868$, respectively). Stone characteristics were also similar in terms of stone size, number, composition and location ($p = 0.488$, $p = 0.157$, $p = 0.823$, $p = 0.396$, respectively). Although the stone CT value, which represents the degree of the stone's hardness, was slightly higher in the SMP group ($878.56 \pm 242.32$) than the F-URS group ($844.94 \pm 274.80$), it did not reach statistical significance ($p = 0.245$).

The main perioperative parameters and outcomes were presented in Table 2. The mean American Society of Anesthesiologists classification score was similar between the groups ($p = 0.279$). The mean operation time was significantly shorter in the SMP group than the F-URS group ($78.85 \pm 18.12$ min, $96.74 \pm 21.10$ min, p <0.001, respectively). The mean postoperative hemoglobin drop was not routinely assessed in the F-URS group unless any severe complications occurred, while it was calculated to be $10.44 \pm 7.92$ g/L in the SMP group ($p < 0.001$). Furthermore, significant differences in mean postoperative stays and overall cost were found between the two groups. Patients who underwent SMP had much longer postoperative hospitalization stays than those who underwent F-URS ($p < 0.001$). However, F-URS cost significantly more than SMP , whatever in patients with single procedure or more ($p < 0.001$; $p < 0.001$). Completely tubeless SMP was performed in 14 (29.17%) patients, whereas all patients in the F-URS group accepted a double-J stent placement. According to the Clavien classification system, no major intraoperative complications (include minimal ureteral perforation) occurred in our study, while postoperative complications were more common in the SMP group ($p < 0.001$). Clavien grades I and II were the most common type of complications that occurred in both groups. Homolateral lumbago and discomfort due to the retention of catheter or nephrostomy tube were the most common Clavien grade 1 complications (7

**Table 1** Comparison of patients, demographic and stone characteristics.

|  | F-URS | SMP | *P* value |
|---|---|---|---|
| **Number of patients** | 104 | 48 |  |
| **Age**[a] **(years)** | 48.72 ± 13.56 | 49.96 ± 12.86 | 0.610 |
| **Gender (Male/Female),** *n* | 71/33 | 34/14 | 0.751 |
| **BMI**[a] **(kg/m$^2$)** | 34.09 ± 2.20 | 33.11 ± 2.17 | 0.458 |
| **Cummulative stone burden**[a]**, mm** | 24.19 ± 3.44 | 24.50 ± 3.61 | 0.488 |
| **Stone CT value**[a]**, Hu** | 844.94 ± 274.80 | 878.56 ± 242.32 | 0.245 |
| **Stone Composition** |  |  |  |
| *Calcium* | 74 | 35 |  |
| *Non-calcium* | 30 | 13 | 0.823 |
| **Operation side(R/L)** | 47/57 | 21/27 | 0.868 |
| **Stone number** |  |  | 0.157 |
| *Multiple* | 50 | 29 |  |
| *Single* | 54 | 19 |  |
| Stone location |  |  | 0.396 |
| *Multiple calyx* | 25 | 17 |  |
| *Renal pelvis* | 50 | 15 |  |
| *Upper calyx* | 6 | 3 |  |
| *Middle calyx* | 5 | 3 |  |
| Lower calyx | 18 | 10 |  |

**Notes.**
F-URS flexible ureteroscopy, SMP super-mini percutaneous lithotripsy, BMI body mass index, CT computed tomography.
[a] Mean.

in the F-URS group and 12 in the SMP group). Postoperative fever requiring antibiotic therapy (according to the urine culture or blood culture; Clavien grade II) was found in 5 patients and 11 patients in the F-URS and SMP groups, respectively. One patient in the SMP group required blood transfusions owing to a significant drop in hemoglobin (Clavien grade II). Thus, the patient received a selective angioembolization after SMP (Clavien grade IIIa). A double-J stent was re-inserted under general anesthesia in 2 patients in the SMP group because of the migration of the former double-J stent (Clavien grade IIIb).

The mean number of procedures was different in these two groups, as the SMP group required significantly fewer stage-2 procedures ($1.28 \pm 0.45$, $1.10 \pm 0.31$, $p = 0.032$). The primary SFR was 47.12% for the F-URS group and 81.25% for the SMP group ($p < 0.001$). However, the SFR increased to 86.54% and 93.75% 3 months later, showing no significant difference between the two groups ($p = 0.190$).

Clinical evidence showed that SFR related to several factors such as stone location and treatment modality (*De et al., 2015*; *Miernik et al., 2012*). In our study, a more effective primary SFR was presented in the SMP group than in the F-URS group when faced with renal stones located in the renal pelvis and lower calyx ($p = 0.011$, $p = 0.009$, Table 3). There was no significant difference in the primary SFR between the two groups for stones located in multiple calyx, middle calyx and upper calyx ($p = 0.051$, $p = 0.595$, $p = 0.303$, Table 3). Moreover, no significant difference in stone location could be detected 3 months later

**Table 2  Comparison of perioperative parameters and outcomes.**

|  | F-URS | SMP | *P* value |
|---|---|---|---|
| **Number of patients** | 104 | 74 | |
| **Number of patients** | 104 | 48 | |
| **ASA score**[a] | 1.88 ± 0.72 | 1.75 ± 0.70 | 0.279 |
| **Operation time**[a]**, min** | 96.74 ± 21.10 | 78.85 ± 18.11 | **<0.001** |
| **Double-J stent placement**, *n* | 104 | 34 | **<0.001** |
| **Hemoglobin drop**[a]**, g/l** | NA | 10.44 ± 7.92 | **<0.001** |
| **Postoperative stay**[a]**, days** | 3.31 ± 1.87 | 5.02 ± 1.54 | **<0.001** |
| **Cost, US Dollar** | | | |
| *Single procedure* | 6417.13 ± 266.33 | 4343.94 ± 425.25 | **<0.001** |
| *Two procedures* | 13084.83 ± 472.82 | 8179.48 ± 344.83 | **<0.001** |
| **Complication rate,%** | 11.54%(12) | 56.25%(27) | **<0.001** |
| *Grade I* | 7 | 12 | **0.002** |
| *Grade II* | 5 | 12 | **<0.001** |
| *Grade IIIa* | 0 | 1 | 0.14 |
| *Grade IIIb* | 0 | 2 | **0.036** |
| **Number of procedures**[a] | 1.28 ± 0.45(29) | 1.10 ± 0.31(5) | **0.032** |
| **SFR, %** | | | |
| *Primary* | 47.12%(49) | 81.25%(39) | **<0.001** |
| *3 months later* | 86.54%(90) | 93.75%(45) | 0.190 |

**Notes.**

F-URS flexible ureteroscopy, SMP super-mini percutaneous lithotripsy, ASA score American Society of Anesthesiologists, SFR stone free rate.

[a]Mean.

for SFR between the two groups ($p = 0.080$, $p = 0.666$, $p = 0.296$, $p = 0.554$, respectively, Table 3).

## DISCUSSION

Currently, the preferred treatment modalities for nephrolithiasis include extracorporeal shock wave lithotripsy (ESWL), F-URS and PCNL. The European Association of Urology (EAU) guidelines recommend ESWL as one of the first-line treatment for urolithiasis <20 mm. However, the SFR of ESWL decreases to 50% in cases with stones larger than 20 mm or multiple renal stones (*Türk et al., 2016*). In obese patients, a frequent limiting factor for the effectiveness of ESWL is an inability to position the patient to allow the stone to be within the focal point of the lithotripter. Great machines such as the Dornier HM3 or MPL9000 help with this to some extent, but they do not ensure the success of the procedure (*Yang & Bellman, 2004*). Therefore, F-URS and PCNL become a more favorable option for obese patients with renal stones larger than 20 mm.

The application of F-URS has been widespread for the treatment of intra-renal stones due to its many recognized advantages, including: minimal invasiveness, less blood loss and fewer post-operative hospital stays (*Doizi et al., 2015*; *Xu et al., 2015*; *Wang et al., 2018*). However, the high cost of F-URS including operation-related cost for patients and the original purchasing cost, the reprocessing cost and repair fees of the flexible ureteroscope

**Table 3  Comparison of primary and 3 months later SFR.**

|  | F-URS | SMP | *P* value |
|---|---|---|---|
| **Primary SFR** | | | |
| *Multiple calyx* | (10/25) | (12/17) | 0.051 |
| *Renal pelvis* | (25/50) | (13/15) | **0.011** |
| *Upper calyx* | (3/6) | (2/3) | 0.595 |
| *Middle calyx* | (4/5) | (3/3) | 0.303 |
| *Lower calyx* | (7/18) | (9/10) | **0.009** |
| **3 months later** | | | |
| *Multiple calyx* | (16/25) | (15/17) | 0.080 |
| *Renal pelvis* | (48/50) | (14/15) | 0.666 |
| *Upper calyx* | (5/6) | (3/3) | 0.296 |
| *Middle calyx* | (5/5) | (3/3) | / |
| *Lower calyx* | (16/18) | (10/10) | 0.554 |

**Notes.**
F-URS flexible ureteroscopy, SMP super-mini percutaneous lithotripsy, SFR stone free rate.

for healthcare institutions limit the use of this procedure worldwide (*Schoenthaler et al., 2015*). Additionally, small residual fragments after F-RUS need to be evacuated by the patients themselves via certain activities such as jumping and handstands, which was quite difficult for many obese patients, increasing the risk of stone recurrence (*Iremashvili et al., 2018*).

Both EAU and AUA (American Urological Association) guidelines recommend PCNL as the first-line therapy for renal stones larger than 20 mm (*Türk et al., 2016*). When compared with ESWL or F-URS, conventional PCNL is usually associated with higher SFR and complication rates (*De et al., 2015*). Recent studies have confirmed that many complications of PCNL are associated with percutaneous access, and the size of PCNL access tracts was considered a major contributing factor (*Liu et al., 2018*; *Zhu et al., 2015*). SMP is a novel technique and instrumentation for PCNL, used to improve the safety and efficacy of PCNL. *Zeng et al. (2016)* reported their successful experience with SMP and consider the procedure to be a safe and effective treatment for renal stones <25 mm. Their later studies showed that SMP was equally effective as Miniaturized PCNL (MPCNL) while it possessed significant advantages for hospital stay lengths and tubeless rates (*Liu et al., 2018*). However, in obese patients, the application of SMP still remains a challenge not only because of the increased risk of anesthesia, but also the inefficient ultrasonographic visualization of the target pelvicalyceal system and stones due to excess fat tissue (*Streeper et al., 2016*). Furthermore, the Fr7 sheath and Fr4.5 nephroscope of SMP improve the flexibility of the procedure. Yet it was also easily snapped, especially in obese patients. Enhanced SMP with a Fr11 sheath may help solve the problem.

In our study, significant difference in mean operation time was found between the SMP group and the F-URS group. However, in Chen's study, the mean operation time of MPCNL was similar to the F-URS (*Chen et al., 2018*), while *Ozgor et al. (2016)* and *Ozgor et al. (2018)* reported the mean operation time of MPCNL was significantly longer than the F-URS in their two studies. This might be related to the particular way the stone fragments

were managed by the SMP. In the SMP procedure, the stone fragments were evacuated spontaneously through the negative pressure aspiration. This reduces intra-renal pressure, so that forceps or graspers are not required for the retrieval of stone fragments in most cases. Furthermore, 14 patients accepted total tubeless SMP in the present study, which might also contribute to the decrease of the operation time.

When compared with PCNL, F-URS was widely recognized for its faster post-operative recovery (*Xu et al., 2015*; *Miernik et al., 2012*). In our study, the mean post-operative duration of SMP was also longer than the F-RUS, as SMP still made certain wounds to the target kidney and the patients needed more recovery time in bed. Meanwhile, a nephrostomy tube was placed in most cases of SMP patients, which also needs time to be removed. Comparatively higher complication rates in the SMP group might also contribute to the longer post-operative stays of SMP patients. However, we thought the post-operative stay could be shortened if most SMPs could be completed as an outpatient procedure. However in China, certain hospital costs can be reimbursed according to the social insurance policy but outpatient expenses are not covered. The current medical environment in our country is poor, therefore it was not easy to convince the renal stone patients to accept the SMP procedure in outpatient.

SFR is probably the most important clinical parameter for evaluating the efficacy of a lithotripsy technique. In our study, 29 patients in the F-URS group and 5 patients in the SMP group required an auxiliary procedure to further disintegrate the residual stone fragments ($p = 0.032$). With regard to the SFR, significant differences were found between the two groups on post-operative 1 day, although no differences were found at the 3 month follow-up, regardless of stone location. Our results were similar to the study reported by *Chen et al. (2018)*, who found that the SFR for F-URS was significantly lower than for MPCNL after the first session. However, *Ozgor et al. (2016)* and *Ozgor et al. (2018)* reported in their two studies that on both post-operative 1 day and at a 3 month follow-up, the two groups showed no significant differences between them. We suggested this may be a benefit from using a narrow caliber nephroscope and metal-access sheath, which may help facilitate the reach to target calyces. Additionally, the negative pressure aspiration system used in SMP may also contribute to the extraction of stone fragments during the operation.

In the present study, significant differences were found between the two groups in the occurrence of post-operative complications. The overall number of complications was lower in the F-URS group than in the SMP group (11.54% versus 56.25%, $p < 0.001$). More than twice the number of patients suffered a fever after SMP than those following F-URS. We believe this may be due to a block in the negative pressure aspiration tract caused by certain fragments intra-operatively. The blockage caused high transitional pressure in the collecting system, which may consequently lead to pyelovenous or pyelosinus backflow, finally increasing the overall risk of post-operative fever. Two patients in the SMP group suffered a double J stent migration after the operation and a further cystoscopy was performed to readjust the position of the stent under general anesthesia. In several published studies, the rate of blood transfusion after percutaneous stone extraction ranged from 0% to 45% (*Ganpule, Shah & Desai, 2014*). In our study, only one patient accepted a

blood transfusion post-operatively due to a significant drop in hemoglobin, and later the patient also accepted a selective angioembolization. This may benefit from the application of the reduced diameter of percutaneous tract.

Today, physicians face growing pressure for cost-effectiveness from healthcare providers and clinical institutions. F-URS is an expensive procedure which uses endoscopes with a short lifetime cycle and necessary, high-priced disposables. Compared to F-URS, ultra-mini PCNL (UMP) presented significantly lower costs for endoscopes and disposables (*Schoenthaler et al., 2015*). Similarly, the mean overall cost for a single procedure of F-URS in our center was nearly 7,000 US dollars, which was more expensive than SMP. This may because the F-URS procedure requires certain high-priced disposables such as nitinol baskets, ureteral access sheaths and guide wires. Furthermore, additional ancillary procedures of F-URS may be required for patients with larger stones.

Although our study is the first investigation to focus on this topic, certain limitations should be acknowledged. Firstly, the number of patients was small. Secondly, the retrospective nature of the present study was also small. Thirdly, the number of tubeless SMP was limited as the surgeon preferred to place a nephrostomy tube to prevent post-operative infection. In addition, some patients preferred SMP because of lower cost and higher primary SFR, while other patients preferred F-URS due to its reduced invasiveness and lower complication rate. Finally, the use of KUB instead of CT scans in the evaluation of post-operative SFR may have limited.

## CONCLUSION

This study demonstrated that SMP is as effective as F-URS in treating obese patients with 20–30 mm renal stones. However, F-URS offers advantages with a lower complication rate, while SMP performed better in terms of operation time, tubeless rate and overall costs. Nevertheless, our study should be supported by additional prospective randomized studies with larger patient populations.

### Funding

This work was supported by the First Affiliated Hospital of Guangzhou Medical University 2010A060801016 (to Ri-jin Song), the Talent Health Youth Project of Suzhou City GGRC052 and the Youth Project of Suzhou City (kjxw2018073) (to Chen Xu). The funders had no role in study design, data collection and analysis, decision to publish, or preparation of the manuscript.

### Grant Disclosures

The following grant information was disclosed by the authors:
First Affiliated Hospital of Guangzhou Medical University: 2010A060801016.
Talent Health Youth Project of Suzhou City: GGRC052.
Youth Project of Suzhou City: kjxw2018073.

## Competing Interests

The authors declare there are no competing interests.

## Author Contributions

- Chen Xu analyzed the data, prepared figures and/or tables, authored or reviewed drafts of the paper, and approved the final draft.
- Rijin Song conceived and designed the experiments, analyzed the data, prepared figures and/or tables, authored or reviewed drafts of the paper, and approved the final draft.
- Pei Lu analyzed the data, authored or reviewed drafts of the paper, and approved the final draft.
- Minjun Jiang conceived and designed the experiments, prepared figures and/or tables, and approved the final draft.
- Guohua Zeng conceived and designed the experiments, authored or reviewed drafts of the paper, and approved the final draft.
- Wei Zhang conceived and designed the experiments, performed the experiments, authored or reviewed drafts of the paper, and approved the final draft.

## Human Ethics

The following information was supplied relating to ethical approvals (i.e., approving body and any reference numbers):

The ethical committee of the Ninth People's Hospital of Suzhou granted ethical approval to carry out the study within its facilities (Ethical Application Ref: 19-01).

## Data Availability

Raw data is available in the Supplemental Files.

## Supplemental Information

Supplemental information for this article can be found online at http://dx.doi.org/10.7717/peerj.8532#supplemental-information.

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
