# Peer review of "A retrospective study comparing super-mini percutaneous nephrolithotomy and flexible ureteroscopy for the treatment of 20–30 mm renal stones in obese patients"

_PeerJ, doi:10.7717/peerj.8532_

## Round 0.1 · original submission · Major Revisions

Interesting study, however please refer to the reviewer comments.
thanks

best regards

Reviewer 1 ·

Basic reporting

3. Flexible ureterocopy was not the first line choice for stone larger than 20mm, why would you compare RIRS with SMP in these patients with such a large stone burden?

Experimental design

it’s a pity that, the present study was a retrospective study, a prospective study was better.
2. The enrolled cases in SMP group was limited, thus there might be some selective bias in the present study, please enroll more cases in SMP group.

Validity of the findings

The results in table 2 was not completed.

Additional comments

1. Interesting study, however, it’s a pity that, the present study was a retrospective study, a prospective study was better.
2. The enrolled cases in SMP group was limited, thus there might be some selective bias in the present study, please enroll more cases in SMP group.
3. Flexible ureterocopy was not the first line choice for stone larger than 20mm, why would you compare RIRS with SMP in these patients with such a large stone burden?
4. Why the patients with body mass index (BMI)>30kg/m were concerned in the present study? Was there any special issue in SMP or RIRS in these patients?
5. What’s your selection creteria for SMP and RIRS in these stone larger than 20mm?
6. A KUB radiogram was performed 3 days postoperatively to assess the primary SFR in SMP group, how about the patients with tubeless-SMP? I think it would contribute to a longer hospital stay in SMP group when compared to RIRS.
7. Why patients received RIRS stayed in hospital for 2.8 days after the operation?
8. The results in table 2 was not completed.
9. The discussion was not well presented.
10. Crammer and spelling mistakes should be revised by native speaking guys.

Reviewer 2 ·

Basic reporting

This is an interesting topic in endourology, in which less invasive approaches in stone disease is studied to improve outcomes, lower complication rates and improve future techniques with the consideration to the costs at the same time also.
The authors are comparing between two procedures one that is widely used and another as a new alternative and less invasive to the traditional PCNL.
The overall English is good but needs to be revised and proofread in areas.

Experimental design

Single senior surgeon with patients
Few questions rise for the methods used and few recommendations:

1- Please check and explain 106-108, you mention use of semi-rigid ureteroscope in a SMP procedure?.
2- Were patients counseled only to choose between 2 procedures or not? and is this a routine way to treat stones of 20-30mm with FURS or SMP at your institution?
3- I recommend mentioning stone composition in addition to mentioning hardness of the stone by HU. That would add validity to your results.

Validity of the findings

The study does show interesting findings, few points:
1- In table 2, you mention that 22 patients out of 37 of the SMP group had complications, that does not reflect the number you give of 29.73%. On the other hand FURS had 12 patients out of 104 (11.54%)
2- Please stratify by type of complication and compare between both groups to give a better understanding what are the more to less common complications in each group.
3- In the discussion, please discuss further about the reason postop stay was longer in SMP.

Additional comments

no comment

---

## Round 0.2 · Major Revisions

Dear authors,

Please revise the paper according to the reviewer comments.

Thanks

Reviewer 1 ·

Basic reporting

Grammer and spelling still require major revison.

Experimental design

The sample size in SMP group was a little bit small.
Retrospective study had potential flaws.

Validity of the findings

NA

Additional comments

I did not get the response for my previous question. Please send it to the editors and transfer to me, thanks.
1. Interesting study, however, it’s a pity that, the present study was a retrospective study, a prospective study was better.
2. The enrolled cases in SMP group was limited, thus there might be some selective bias in the present study, please enroll more cases in SMP group.
3. Flexible ureterocopy was not the first line choice for stone larger than 20mm, why would you compare RIRS with SMP in these patients with such a large stone burden?
4. Why the patients with body mass index (BMI)>30kg/m were concerned in the present study? Was there any special issue in SMP or RIRS in these patients?
5. What’s your selection creteria for SMP and RIRS in these stone larger than 20mm?
6. A KUB radiogram was performed 3 days postoperatively to assess the primary SFR in SMP group, how about the patients with tubeless-SMP? I think it would contribute to a longer hospital stay in SMP group when compared to RIRS.
7. Why patients received RIRS stayed in hospital for 2.8 days after the operation?
8. The results in table 2 was not completed.
9. The discussion was not well presented.
10. Crammer and spelling mistakes should be revised by native speaking guys.

---

## Round 0.3 · accepted · Accept

Well performed review, which shows the real value of your interesting study.

Reviewer 1 ·

Basic reporting

The revison had great improvement than the previous one in language and spelling, it could be better after native speaking editor's revision.
In the introduction, the indications for RIRS and SMP should be clarified, why 2-3cm stone received RIRS? Why obese patients were selected in this study?

Experimental design

The sample size should be clarified.

Validity of the findings

The detailed complications in both groups should be presented, as well as the total SFR.